# JAK/STAT signaling in *Drosophila* muscles controls the cellular immune response against parasitoid infection

Hairu Yang[1], Jesper Kronhamn[1], Jens-Ola Ekström[1,2], Gül Gizem Korkut[1,†] & Dan Hultmark[1,2,*]

## Abstract

The role of JAK/STAT signaling in the cellular immune response of *Drosophila* is not well understood. Here, we show that parasitoid wasp infection activates JAK/STAT signaling in somatic muscles of the *Drosophila* larva, triggered by secretion of the cytokines Upd2 and Upd3 from circulating hemocytes. Deletion of *upd2* or *upd3*, but not the related *os* (*upd1*) gene, reduced the cellular immune response, and suppression of the JAK/STAT pathway in muscle cells reduced the encapsulation of wasp eggs and the number of circulating lamellocyte effector cells. These results suggest that JAK/STAT signaling in muscles participates in a systemic immune defense against wasp infection.

**Keywords** *Drosophila*; innate immunity; JAK/STAT signaling; muscles
**Subject Categories** Immunology; Microbiology, Virology & Host Pathogen Interaction

## Introduction

When infected, *Drosophila melanogaster* activates humoral as well as cellular immune responses [1–5]. Best understood is the humoral immune response to bacterial or fungal infections, which leads to the production of several antimicrobial peptides. This response is mainly controlled by two NF-κB-like signaling pathways, the Toll and the immune deficiency (IMD) pathways [6–8], with somewhat different specificities toward different microorganisms [2,3]. The cellular immune responses are more complex, involving phagocytosis of invading microorganisms or encapsulation of larger parasites. Two classes of blood cells, or hemocytes, are present in healthy larvae: plasmatocytes, which are phagocytically active, and crystal cells, which deposit melanin around wound sites [9]. In response to infection by parasitoid wasps, such as *Leptopilina boulardi,* some plasmatocytes differentiate to generate a third type of hemocytes,

the lamellocytes [4,10]. At least two of these hemocyte classes participate in the encapsulation of the wasp egg. First, plasmatocytes recognize and bind to the invading wasp egg. Then, lamellocytes form a dissociation-resistant layer next to the primary plasmatocyte layer, the capsule. Finally, components of the phenol oxidase cascade, possibly from the crystal cells but more likely from the lamellocytes [11], cause melanization of the wasp egg.

A phenotype akin to the encapsulation response can be found in certain *Drosophila* mutants, with increased numbers of circulating hemocytes, including lamellocytes, and with hemocytes that aggregate in melanized masses, so-called melanotic nodules (or melanotic "tumors") [12]. For instance, melanotic nodules are observed in gain-of-function mutants with constitutively activated JAK/STAT (Janus kinase/signal transducer and activator of transcription) or Toll signaling [13–16]. Several signaling pathways, including JAK/STAT, Toll, JNK, and Rac also generate a similar phenotype when they are specifically activated in the hemocytes [17]. However, the role of these signaling pathways in the response to a parasite infection is not clear. Sorrentino *et al* showed that loss-of-function mutants in the JAK/STAT and Toll pathways have a reduced capacity to encapsulate eggs of *Leptopilina boulardi* [18], suggesting that these pathways are involved in this response. Furthermore, Williams *et al* could link Rac and JNK signaling in hemocytes to the activation of these cells [19–21].

Here, we have investigated the specific role of JAK/STAT signaling in the encapsulation response. In *Drosophila*, the JAK/STAT pathway is relatively simple, with a single cytokine class I receptor, Domeless (Dome) [22], a single JAK homolog, the tyrosine kinase Hopscotch (Hop) [23], and a single STAT transcription factor, Stat92E [24,25]. Only three cytokine-like ligands are so far known to interact with Domeless in *Drosophila*: Outstretched (Os, also called Unpaired, Upd), Unpaired 2 (Upd2), and Unpaired 3 (Upd3). For simplicity, we will here refer to them as Upd1, 2 and 3. While the core components of the JAK/STAT signaling pathway are well conserved between insects and vertebrates, the three *Drosophila* ligands are more divergent [26].

The three Unpaired ligands can bind to the receptor Domeless [27], leading to recruitment and phosphorylation of the JAK

1 Department of Molecular Biology, Umeå University, Umeå, Sweden
2 Institute of Biomedical Technology, BMT, Tampere University, Tampere, Finland
*Corresponding author. Tel: +46 90 785 67 78; E-mail: dan.hultmark@ucmp.umu.se
†Present address: Department of Comparative Physiology, Evolutionary Biology Center, Uppsala University, Uppsala, Sweden

 

homolog Hop. Thereafter, activated Hop phosphorylates Stat92E, a homolog of the mammalian STATs. Finally, activated Stat92E translocates into the nucleus and induces various target genes, which exert different effects on the cells, depending on the tissue or cell context, including proliferation, differentiation, migration, apoptosis, and cell survival [28]. Surprisingly, JAK/STAT signaling is known to suppress hematopoiesis in *Drosophila* [29], in apparent contradiction to the observed requirement for this pathway in the cellular immune response.

In this study, we found that JAK/STAT signaling in somatic muscles plays an important role in the response of *Drosophila* larvae against wasp infection, besides its role in hemocytes and hematopoietic tissue. The presence of a wasp egg activates JAK/STAT signaling in muscles, induced by Upd2 and Upd3 secretion from hemocytes. Suppression of JAK/STAT signaling in muscles seriously reduces the immune response against wasp infection.

## Results

### JAK/STAT pathway activation upon wasp infection

To follow JAK/STAT pathway activity in living *Drosophila* larvae upon wasp infection, we used animals that carried the JAK/STAT GFP reporter, *10xStat-GFP* [30]. Twenty-six hours after infection with *Leptopilina boulardi* wasp eggs, we observed strong induction of GFP expression in the infected larvae. Surprisingly, the induced GFP expression was primarily located to the somatic muscles (Fig 1A–C). To confirm this observation, we also assayed the expression of the STAT-inducible *Socs36E* gene, a negative regulator of JAK/STAT signaling. The expression of this gene increased

approximately twofold in a muscle preparation after wasp infection (Fig 1D). The response is relatively slow; no obvious increase in 10xStat-GFP fluorescence was detected 4 h after infection, but at 8 h a significant effect could be observed (Appendix Fig S1A–D).

Because circulating hemocytes are difficult to visualize in living larvae, we also investigated hemolymph samples separately and found increased GFP expression in plasmatocytes but not in lamellocytes after wasp infection (Fig 1, compare panels E, E′ to F, F′). Thus, wasp infection induces JAK/STAT activation both in larval plasmatocytes and in somatic muscles. The strongest response is observed in muscles, but we cannot exclude weaker responses in other tissues. The fat body is largely negative, although the GFP reporter is sometimes activated in small regions of this tissue after wasp infection (Appendix Fig S1E and F).

### JAK/STAT, P38, or JNK activation in hemocytes is sufficient but not required to activate JAK/STAT signaling in somatic muscles

Activation of JAK/STAT, P38, or JNK signaling in circulating hemocytes triggers the generation of lamellocytes, which appear in the circulation [17]. We therefore investigated whether JAK/STAT signaling is activated in the muscles under these conditions. We used a hemocyte-specific driver, *Hemolectin-GAL4* (*Hml-GAL4*) [31], to individually activate the JAK/STAT, P38, or JNK pathways, using the *UAS-hop^{Tum}*, *UAS-Mekk1*, or *UAS-hep^{CA}* constructs, respectively. As shown in Fig 2A–E, activation of any of these pathways in hemocytes induced JAK/STAT signaling in muscles, indicating that hemocytes send out signals that activate JAK/STAT signaling in the muscles. Thus, activation of any one of the JAK/STAT, P38, or JNK pathways in hemocytes is sufficient to activate the JAK/STAT pathway in larval muscles.

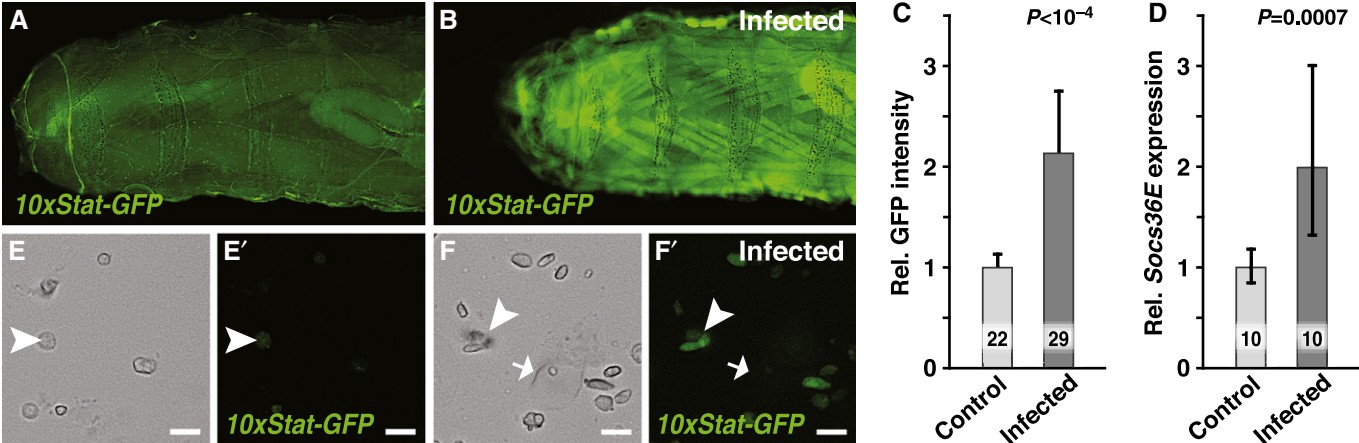

**Figure 1.  Activation of JAK/STAT signaling in skeletal muscles by wasp infection.**

A, B    JAK/STAT signaling was detected with the *10xStat-GFP* reporter in (A) uninfected control larvae, and (B) larvae 27 h after wasp infection.

C    Quantification of GFP signal in muscles in the indicated total number of larvae from three independent experiments. Uninfected control = 1. Bars show average and standard deviation. The *P*-value for significant difference from the uninfected control is indicated above (unpaired *t*-test).

D    Gene *Soc36e* expression in muscles was assayed by quantitative PCR before or after wasp infection. Uninfected control = 1. Bars show average from five independent experiments, and the error bars indicate the span calculated from ± 1 standard deviation of the normalized $C_t$ values. The *P*-value for significant difference from the uninfected control is indicated above (unpaired *t*-test).

E–F′    Bright-field and *10xStat-GFP* fluorescence images of plasmatocytes (arrowheads) and lamellocytes (arrows) from (E, E′) control larvae, and (F, F′) larvae 27 h after wasp infection. Scale bars: 10 μm.

Next, we asked whether activation of these pathways is also necessary for the JAK/STAT response in larval muscle cells after wasp infection. Similar strategies as above were used to suppress these pathways in hemocytes: $UAS\text{-}dome^{DN}$ for the JAK/STAT pathway, $UAS\text{-}P38b^{DN}$ for the P38 pathway, and $UAS\text{-}bsk^{DN}$ for the JNK pathway. We found that none of these constructs significantly inhibit JAK/STAT activation in muscles upon wasp infection (Fig 2F–J), suggesting that none of these pathways are required in hemocytes for JAK/STAT activation in muscles. However, we cannot exclude the possibility that the pathways act redundantly.

## Upd2 and Upd3 from hemocytes induce JAK/STAT pathway activation in somatic muscles

We then asked whether one or more of the cytokines Upd1–3 might act as signals to the muscles, and from which tissues they are secreted. For that purpose, we first used *upd1-GAL4* and *upd3-GAL4* stocks, coupled to a *UAS-GFP* reporter, to visualize the *upd1* and *upd3* expression patterns. A corresponding reporter for *upd2* was not available. We found that the *upd3* reporter was strongly induced in plasmatocytes after wasp infection (Fig 3, compare panels C and D) but not in other tissues (Appendix Fig S2C and D). However, we observed no obvious *upd1* induction, neither in hemocytes (Fig 3, compare panels A and B) nor in other tissues after wasp infection (Appendix Fig S2A and B). To confirm this observation, and to investigate the possible role of *upd2*, we used quantitative RT–PCR to assay *upd1*, *upd2* and *upd3* transcripts both in hemocytes and in the remaining parts of the larval body after flushing out the hemocytes. Our results show that both *upd2* and *upd3* transcripts are dramatically induced in hemocytes after wasp infection, 11-fold for *upd2* and 38-fold for *upd3* (Fig 3E). No significant induction was detected in the remaining tissues of the larva (Fig 3F). Again, *upd1* transcripts were not affected, neither in hemocytes nor in the corpse (Fig 3E and F). Thus, wasp infection causes circulating hemocytes to express and most likely secrete the Upd2 and Upd3 cytokines, but not Upd1.

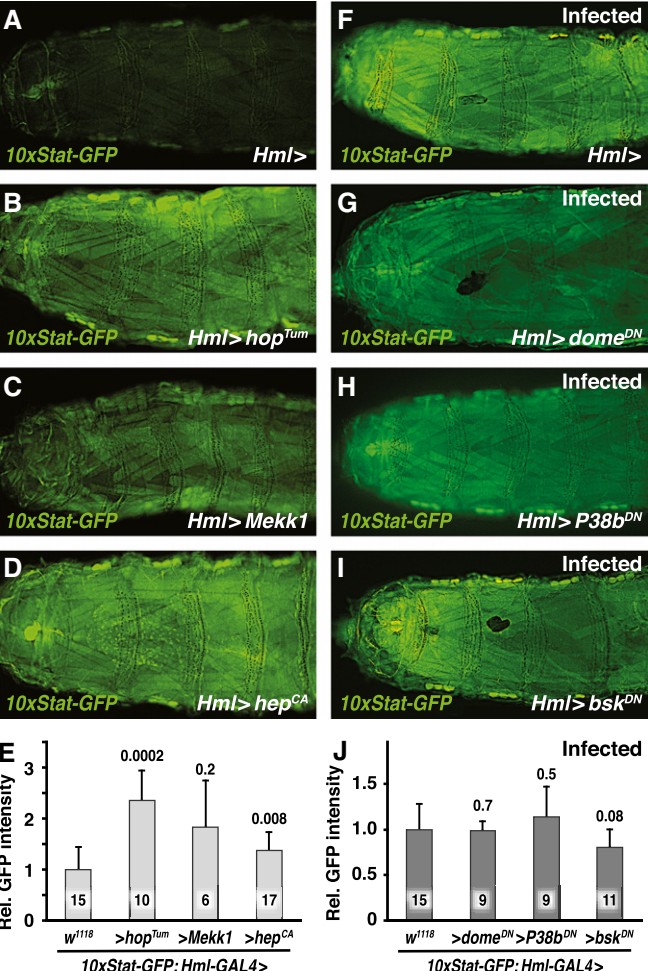

**Figure 2.  Activation of JAK/STAT signaling in skeletal muscles by activation of the JAK/STAT, P38, or JNK pathways in hemocytes.**

A–D  JAK/STAT signaling, as detected with the *10xStat-GFP* reporter, is low in the *Hml* driver control (A), but activated after hemocyte-specific overexpression of $hop^{Tum}$ (B), *Mekk1* (C), or $hep^{CA}$ (D).

E  Quantification of GFP signal in muscles in the indicated total number of larvae from one or two independent experiments. Bars show average and standard deviation. The *P*-values for significant differences from the uninfected controls are indicated above (unpaired *t*-test).

F–I  Wasp-induced JAK/STAT signaling in the muscles of *Hml* driver control larvae (F) is unaffected when the same driver is used to suppress JAK/STAT (G), P38 (H), or JNK signaling (I), in hemocytes.

J  Quantification of GFP signal in muscles in the indicated total number of larvae from one or two independent experiments. Bars show average and standard deviation. The *P*-values for significant differences from the uninfected controls are indicated above (unpaired *t*-test).

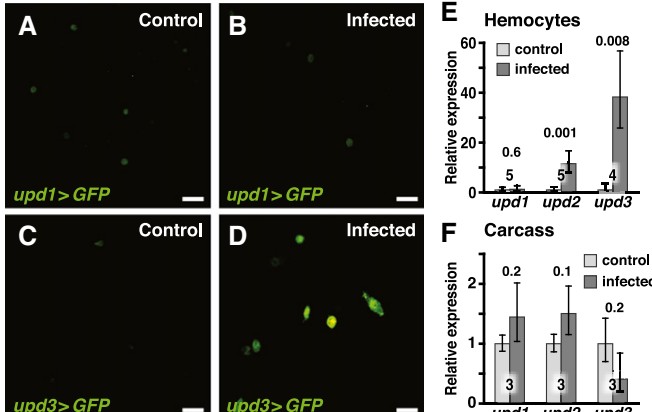

**Figure 3.  The *upd2* and *upd3*, but not *upd1*, cytokine genes are specifically induced in hemocytes after wasp infection.**

A–D  Expression of the *upd1* (A, B) and *upd3* (C, D) genes in hemocytes from control (A, C) and wasp-infected (B, D) larvae, 27 h after infection, as visualized with the *upd1 > GFP* and *upd3 > GFP* reporters. Scale bars: 10 μm.

E, F  Relative expression of the *upd1*, *upd2*, and *upd3* genes were assayed by quantitative PCR in hemocytes (E) or in whole-body larvae after bleeding (F). Uninfected control = 1. Bars show average from three to five independent experiments as indicated, and the error bars show the span calculated from ± 1 standard deviation of the normalized $C_t$ values. The *P*-values for significant differences from the uninfected controls are indicated above (unpaired *t*-test).

To test whether Upd2 or Upd3 expression in hemocytes is sufficient to activate JAK/STAT pathway in somatic muscles, we artificially overexpressed the *upd2* or *upd3* genes in hemocytes using the hemocyte-specific *Hml-GAL4* driver. As expected, GFP expression from the *10xStat-GFP* reporter was strongly induced in the somatic muscles under these conditions (Fig 4A–D, compare panels B and C to panel A), indicating that Upd2 and Upd3 from hemocytes can cell non-autonomously activate JAK/STAT signaling in somatic muscles.

In agreement with a direct role for the Upd2 and Upd3 cytokines in the activation of JAK/STAT in muscles, the homozygous single

mutants *upd2^A* and *upd3^A* [32] both show reduced JAK/STAT activation after wasp infection. JAK/STAT activation is partly suppressed in the *upd2^A* mutant and almost completely in *upd3^A* and in the double mutant *upd2^A upd3^A* (Fig 4E–I). We confirmed this observation by using RNAi to suppress *upd2* or *upd3* in hemocytes, using the hemocyte-specific *Hml-GAL4* driver. The wasp-induced JAK/STAT activity in muscles was significantly reduced by suppression of either gene (Fig 4J). These results indicate that Upd2 and Upd3, produced in hemocytes of infected larvae, act additively or synergistically to activate JAK/STAT signaling in somatic muscles.

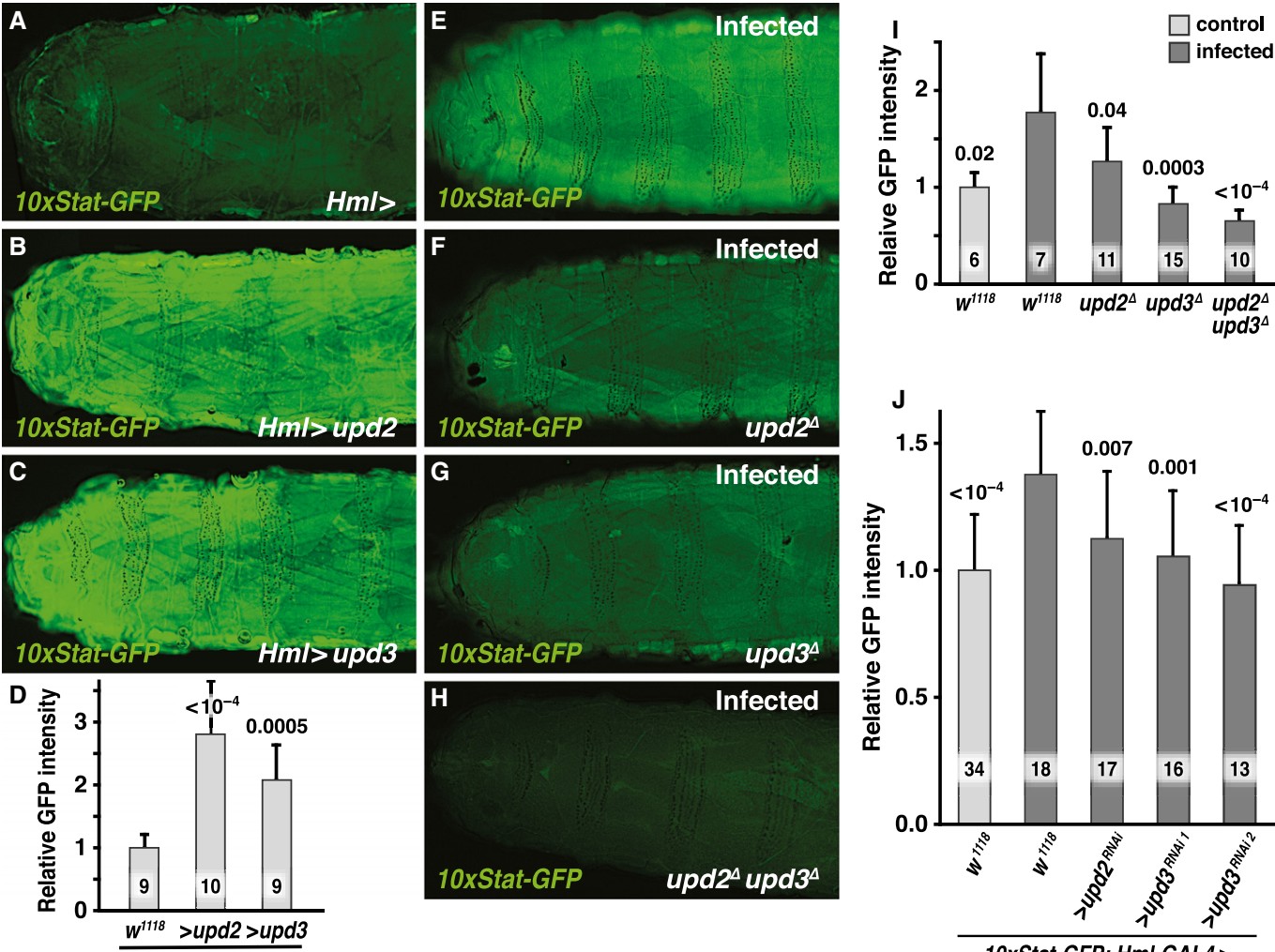

**Figure 4.    Upd2 and Upd3 from hemocytes mediate the wasp-induced JAK/STAT activation in muscles.**

A–C    Hemocyte-specific overexpression of *upd2* (B) or *upd3* (C) with the *Hml-GAL4* driver strongly induces GFP expression in larval muscles, as detected by the *10xStat-GFP* reporter, compared to the wild-type control (A).

D    Quantification of GFP signal in muscles in the indicated total number of larvae in one experiment. Bars show average ± standard deviation. The *P*-values for significant difference from the wild-type control are indicated above (unpaired *t*-test).

E–H    Compared with infected control (E), the male homozygous deletion mutants *upd2^A* (F) and *upd3^A* (G) suppress JAK/STAT activation in muscles. The double homozygous mutant *upd2^A upd3^A* (H) has the strongest effect.

I    Quantification of GFP signal in muscles in the indicated total number of larvae in one experiment. Bars show average ± standard deviation. The *P*-values for significant difference from the infected wild-type control are indicated above (unpaired *t*-test).

J    Wasp-induced JAK/STAT activity in muscles is reduced when *upd2* or *upd3* is suppressed in hemocytes. Quantification of GFP signal in the indicated total number of larvae from two or three independent experiments. Bars show average and standard deviation. The *P*-values for significant difference from the infected wild-type control are indicated above (unpaired *t*-test).

### Role of Upd2 and Upd3 during wasp infection

To investigate the role of Upd2 and Upd3 in the immune response against wasp infection, we let wasps infect the single homozygous mutants $upd2^A$ or $upd3^A$, or the double mutant $upd2^A$ $upd3^A$. After 27 h, we calculated the encapsulation rate, that is the percentage of infected larvae with melanized wasp eggs. The encapsulation rates in both single mutants as well as in the double mutant were significantly reduced to an average of < 10% of the larvae with encapsulated eggs, compared to 55% in the control (Fig 5A), showing that full expression of both cytokines, Upd2 and Upd3, is required for a successful immune response against wasp infection.

To investigate the reason for the low encapsulation rate in the $upd2^A$ and $upd3^A$ mutants, we counted the number of plasmatocytes

and lamellocytes after 15 h of wasp infection. Both mutations, alone or in combination, had significant effects on the number of lamellocytes, which was reduced by more than 50% compared to the wild-type control (Fig 5B). By contrast, the number of plasmatocytes was not significantly affected in any of the mutants tested (Fig 5C). Furthermore, artificial expression of either Upd2 or Upd3 in hemocytes is sufficient to induce lamellocyte formation (Fig 5D), whereas the number of plasmatocytes is unaffected, or even reduced (Fig 5E). It should be pointed out that the number of lamellocytes induced by overexpression of either Upd2 or Upd3 alone is very variable and it only reached 15 and 5%, respectively, of the average number of lamellocytes seen in wasp-infected wild-type larvae (Fig 5, compare panels D and B). These results suggest that other factors are important for full activation of the immune response, as seen in wasp infection. In conclusion, Upd2 and Upd3 expression in the hemocytes is required for a full immune response against wasp infection, and overexpression of either one of them is alone sufficient for at least a partial lamellocyte response.

### JAK/STAT signaling in somatic muscles is required for the cellular immune response

To directly investigate the role of muscle JAK/STAT signaling in the immune response, we suppressed individual components of this pathway in larval somatic muscles by expressing dominant-negative constructs of either the cytokine receptor gene *domeless* ($dome^{DN}$) or the single *Drosophila* STAT gene *Stat92E* ($Stat92E^{DN}$), with the *Mef2-GAL4* muscle-specific driver [33]. This driver is expressed in larval muscles, but not for instance in fat body (Appendix Fig S3A, C and D). Suppressing JAK/STAT in somatic muscles in this way, we observed a significantly reduced encapsulation rate, from 30 to 60% encapsulated eggs in the different control groups down to < 5% in $Stat92E^{DN}$- or $dome^{DN}$-expressing larvae (Fig 6A). We observed the same effect when we expressed $Stat92E^{DN}$ with another muscle-specific driver, *twist-GAL4* (expression pattern in Appendix Fig S3B), which reduced encapsulation from 38 to 3% (Fig 6B). However, artificially activating JAK/STAT signaling by expressing wild-type *Stat92E* in somatic muscles had no effect on the encapsulation rate (Fig 6A), although the *10xStat-GFP* reporter was efficiently activated under these conditions (Appendix Fig S4). These results suggest that JAK/STAT activation in muscles is necessary but not sufficient for a normal cellular immune response. By contrast, we observed no effect on the encapsulation response when we suppressed the JAK/STAT pathway in hemocytes, by expressing the $dome^{DN}$ or $Stat92E^{DN}$ dominant-negative constructs with the combined *He-GAL4* [17] and *Hml-GAL4* hemocyte-specific drivers (Fig 6C). When we counted hemocytes 15 h after wasp infection, we found that suppression of JAK/STAT in muscles significantly reduced the number of circulating as well as tissue-associated lamellocytes (Fig 6D, F and G). Surprisingly, overexpression of the *Drosophila* wild-type STAT gene *Stat92E* in muscles also marginally decreased lamellocyte formation after wasp infection (Fig 6D). The number of plasmatocytes was less affected by JAK/STAT signaling in the muscles, except for a modest suppression by the $Stat92E^{DN}$ construct (Fig 6E). Altogether, our results show that JAK/STAT activation in muscles is required for an efficient hemocyte response against *L. boulardi* infection, including lamellocyte formation and encapsulation of the parasite.

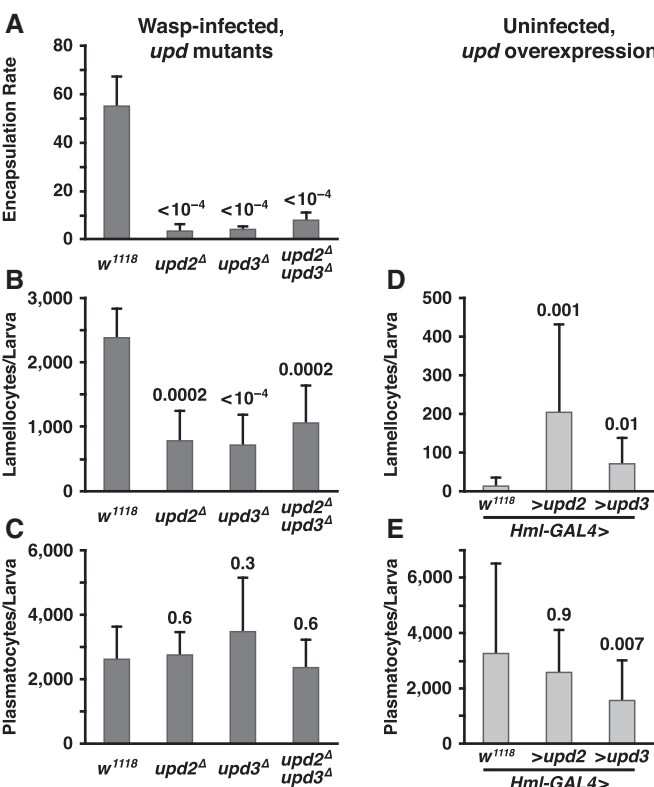

**Figure 5.  Successful encapsulation requires Upd2 and Upd3.**

A   The encapsulation rate is significantly reduced in the homozygous $upd2^A$ or $upd3^A$ mutants, single or combined, compared to the wild-type control $w^{1118}$.

B   In infected larvae, the number of circulating lamellocytes is significantly reduced in all mutants, when assayed approximately 15 h after wasp infection.

C   Plasmatocytes are unaffected in all mutants.

D   In uninfected larvae, the number of lamellocytes is increased when *upd2* or *upd3* is artificially expressed in hemocytes by the *Hml* driver.

E   The number of circulating plasmatocytes is slightly decreased.

Data information: Encapsulation rates (A) were determined in three independent experiments and in total at least 100 larvae were analyzed. Bars show average and standard deviation. The *P*-values (unpaired *t*-test, unequal variance) are indicated. For hemocyte counts (B–E), at least 10 larvae were analyzed for each genotype. Bars show average and standard deviation. The *P*-values (Mann–Whitney test) are indicated.

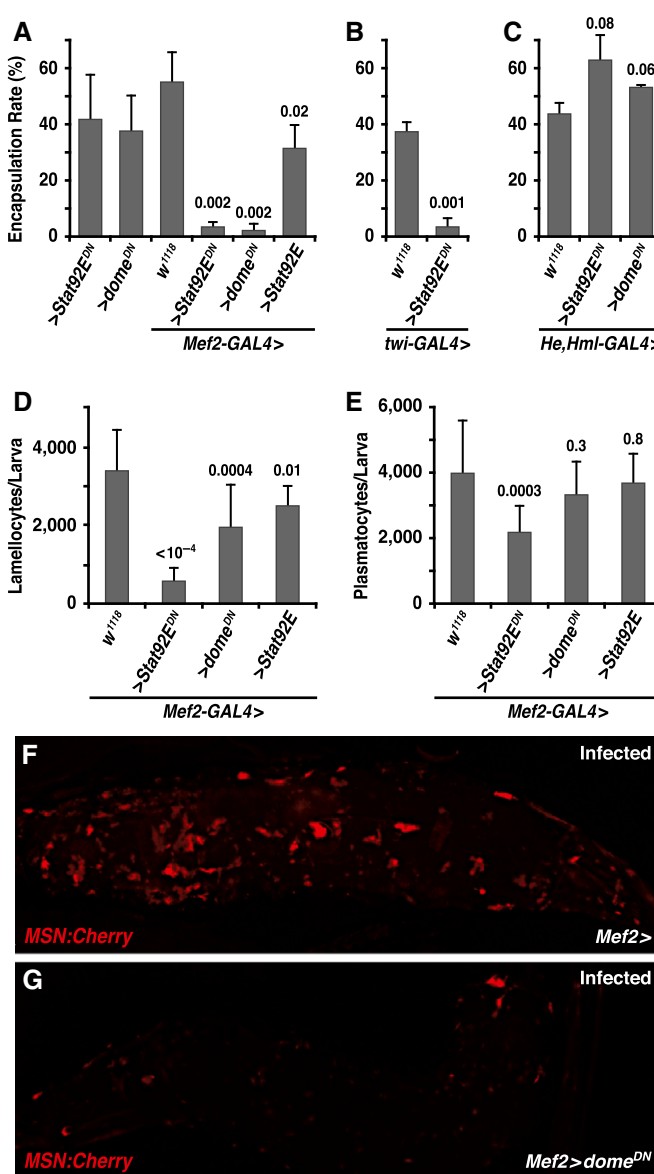

**Figure 6.  JAK/STAT activation in muscles, but not in hemocytes, is required for an efficient cellular immune response.**

A    The encapsulation of wasp eggs is suppressed when dominant-negative JAK/STAT constructs, *Stat92E^DN* or *dome^DN*, are expressed by the muscle-specific *Mef2* driver. Activation of JAK/STAT signaling by overexpression of wild-type *Stat92E* has no effect. Negative controls: the same constructs without driver, and the driver crossed to *w^1118*.

B    The encapsulation of wasp eggs is suppressed when *Stat92E^DN* is expressed by the muscle-specific *twi-GAL4* driver. Negative controls: the driver crossed to *w^1118*.

C    The *Stat92E^DN* and *dome^DN* constructs have no significant effect on encapsulation when they are expressed in hemocytes, using the *Hemese-Hml* double driver.

D, E    The effect of suppressing JAK/STAT signaling in muscles on the number of circulating lamellocytes (D) and plasmatocytes (E).

F, G    Suppression of JAK/STAT signaling in muscles reduces lamellocyte formation. Many *MSN:cherry*-labeled lamellocytes can be seen in control larvae 27 h after wasp infection (F). The number of *MSN:cherry*-labeled lamellocytes was reduced when JAK/STAT signaling was suppressed in muscles with the dominant-negative *dome^DN* construct (G).

Data information: Encapsulation rates (A–C) were determined in three independent experiments, and in total, at least 100 larvae were analyzed. Bars show average and standard deviation. The *P*-values (unpaired *t*-test, unequal variance) are indicated. For hemocyte counts (D, E), at least 20 larvae were analyzed for each genotype. Bars show average and standard deviation. The *P*-values (Mann–Whitney test) are indicated.

## Discussion

The JAK/STAT pathway plays a positive role in the activation of immune responses, both in mammals and in insects. In *Drosophila*, either general upregulation of JAK/STAT signaling by the *hop^Tum* gain-of-function mutant, or cell-specific activation of JAK/STAT in circulating hemocytes, mimics the cellular immune response to wasp infection [14,15,17]. However, although loss-of-function mutants of *hop* or *Stat92E* show significantly decreased encapsulation rates [18], our data show that specific suppression of JAK/STAT in circulating hemocytes does not reduce encapsulation. By contrast, suppression of JAK/STAT in somatic muscles significantly reduces the encapsulation rate and the number of circulating lamellocytes, suggesting that the muscles play a previously unsuspected role in the immune response. Our results show that the activation of muscle cells in turn depends on signals from the hemocytes, in a complex interplay between hemocytes and muscles. Other tissues may also participate in the orchestration of cellular immunity. In the context of hematopoiesis, nerve cells have recently been shown to exert control over the peripheral hemocyte population, in "pockets" between the larval body wall and the somatic muscles [34]. This may also be the site where the feedback between muscles and hemocytes takes place.

The activation of JAK/STAT signaling in somatic muscles, as well as in circulating hemocytes, in infected animals stands in contrast to the lymph gland, a hematopoietic organ where wasp infection is known to suppress the JAK/STAT pathway in the medullary zone [35]. In the lymph gland, JAK/STAT signaling acts to keep hemocyte precursors undifferentiated. Thus, when JAK/STAT signaling is suppressed, the pro-hemocytes of the medullary zone massively differentiate into effector hemocytes [29,35–37]. However, recent work by Minakhina *et al* [38] and Mondal *et al* [39] gives a more complex picture, emphasizing the role of JAK/STAT signaling in the cortical rather than the medullary zone of the lymph gland. They show that constitutive JAK/STAT signaling in cortical cells is cell non-autonomously suppressing the differentiation of neighboring precursor cells, while cell autonomously, it is required for differentiation. Thus, although there is not yet full consensus about the details, the main role of JAK/STAT signaling in the lymph glands is to suppress, rather than activate, hemocyte differentiation. However, a secondary activating role at later stages of differentiation cannot be excluded. Strikingly, flip-out clones that overexpress *UAS-hop* in the lymph gland cortex, thereby activating the JAK/STAT pathway, were found to trigger lamellocyte formation of neighboring cells [38]. The biological role of the latter observation is uncertain, as wasp infection has so far not been reported to activate JAK/STAT signaling in the cortical zone.

Upd1, Upd2, and Upd3 are the only known Domeless ligands, and they are all potentially able to activate the JAK/STAT pathway at long-distance *in vivo* [27,40,41]. Our results show that wasp

infection induces Upd2 and Upd3 expression specifically in hemocytes, while Upd1 is unaffected, indicating that Upd2 and Upd3 are main players in the immune response. This is consistent with the finding of Sorrentino *et al* [18] that the encapsulation of wasp eggs is unaffected in Upd1 (*os⁰/Y*) mutant larvae, while we find that the encapsulation response is strongly impaired in *upd2* or *upd3* null mutants. We were able to mimic the effect of a wasp infection by artificial expression of either Upd2 or Upd3 in circulating hemocytes, leading to dramatic activation of JAK/STAT signaling in muscles and to lamellocyte formation. These results show that hemocytes, via the cytokines Upd2 and Upd3, activate JAK/STAT signaling in the somatic muscles, integrating these tissues in a systemic response against wasp infection. It is not clear why both Upd2 and Upd3 are required for a full response, but it may be related to the different properties of these cytokines. Upd3 is believed to be associated with the extracellular matrix, while Upd2 is freely diffusible [27].

Interestingly, Upd2 and Upd3, but not Upd1, are expressed in the lymph glands under normal conditions [35]. Four hours after wasp infection, Upd3 expression is significantly reduced in the lymph gland, leading to decreased JAK/STAT activity in pro-hemocytes of the medullary zone. This triggers differentiation into mature hemocytes, which aid in the immune response against wasp infection [35]. Again, this illustrates the different and partially opposite roles of JAK/STAT signaling in lymph glands and in circulating hemocytes.

As we have shown, JAK/STAT activation in muscles is not alone sufficient to activate the hemocytes, indicating that the hemocytes must receive additional signals to become activated. Paradoxically however, JAK/STAT activation in the hemocytes is sufficient to trigger lamellocyte formation, although in these cells it is not required.

The direct participation of somatic muscles in the cellular immune response of *Drosophila* was unexpected, but not entirely unprecedented. Jiang *et al* [42] and Buchon *et al* [43] have shown that JAK/STAT signaling is activated in the visceral midgut muscles after a gut infection and that feedback signaling from these muscles controls the regeneration of the midgut epithelium. In mammals, under conditions of chronic inflammation, somatic muscles secrete several inflammatory cytokines, termed myokines, including IL-6, IL-1, IL-8, IL-10, and TNF-α [44], and muscular activity has been demonstrated to influence immune functions in human [45,46]. It is possible that an active participation of muscles in the immune response is a general phenomenon, but we have so far failed to identify a *Drosophila* cytokine that mediates a direct signaling from muscles to hemocytes. An alternative possibility is that the effect of muscles on immunity is indirect, perhaps via redirection of energy resources in the animal.

# Materials and Methods

### Strains

*Drosophila melanogaster* was reared on mashed-potato diet (Appendix Supplementary Methods) at room temperature unless otherwise indicated. *Leptopilina boulardi* G486 wasps [47] were bred on a *D. melanogaster* Canton S stock at room temperature, and

adult wasps were maintained at room temperature in vials with apple juice agar. The following *D. melanogaster* strains were used: *10XSTAT92E-GFP#1* (BL26197) [30], *UAS-Stat92E*^DN^ (*UAS-ΔNStat92E*) [48], *UAS-hop*^Tum^ [15], *UAS-Mekk1* [49], *UAS-dome*^DN^ (*UAS-dome*^ΔCYT^) [22], *UAS-p38b*^DN^ [50], *UAS-hep*^CA^ [51], *UAS-bsk*^DN^ [51], *UAS-upd2* [42], *UAS-upd3* [27], *os-GAL4 > UAS-GFP* [32], *upd3-GAL4 > UAS-GFP* [52], *upd2*^A^ [32], *upd3*^A^ [32], *upd2*^A^ *upd3*^A^ [32], *UAS-upd2*^RNAi^ (BL33949), *UAS-upd3*^RNAi^ (BL28575), *UAS-upd3*^RNAi^ (BL32859), *UAS-Redstinger* (BL8547, here called RFP), *UAS-2EGFP* (BL6874), and *MSNF9mo-mCherry* (here called msn-Cherry) [53]. The following Gal4 driver stocks were used: *Mef2-Gal4* [54] and *Twist-GAL4* (BL2517) [55] are muscle-specific, and *Hemese-Gal4* (*He-Gal4*, BL 8699) [17] and *Hml-GAL4* (BL30139) [56] are hemocyte-specific. Their tissue specificities were tested by crossing to UAS-GFP (Appendix Fig S3). Many stocks are available from the Bloomington Stock Center at Indiana University (BL numbers), others from the laboratories where they were created.

### Encapsulation rate assay

Eggs were collected for 24 h at 25°C, and the eggs were then shifted to 29°C. Once the majority of larvae had developed into the second instar, *L. boulardi* G486 wasps were allowed to lay eggs in the larvae during 2 h at 29°C. The ratio of wasps to *Drosophila* larvae was 1/10. After an additional 26 h, the larvae were washed in phosphate-buffered saline (PBS; 137 mM NaCl, 2.7 mM KCl, 10 mM Na$_2$HPO$_4$ and 2 mM KH$_2$PO$_4$, pH 7.4) and sorted under stereomicroscope according to the presence or absence of black capsules. Larvae without obvious black capsules were dissected to check whether they were infected. Finally, we calculated the encapsulation rate, defined as the ratio of larvae with black capsule to the total number of infected larvae.

### Immobilization of larvae and imaging

Larvae were washed in chilled PBS to remove food and debris, dried with soft tissue, and mounted in 50% glycerol on object slides. The slides were kept at −20°C for 18 min, in order to immobilize the larvae for photography. Immobilized larvae or hemocytes were examined under a NIKON 90i microscope, and images were captured using a NIKON DSFi1 camera and Nis Elements AR software. Software ImageJ was used to quantify these images. The collected and normalized quantified values do not distribute significantly different from the normal distribution (as tested by the D'Agostino-Pearson omnibus normality test, $P = 0.1359$, $n = 175$) and we could therefore use the *t*-test, with Welch's correction for unequal variances, for significance testing.

### Hemocyte counting

Hemocyte counting was performed as previously described [17]. Briefly, 15 h after wasp infection, larvae were washed in PBS, ripped open, using watchmakers' forceps, and bled into 20 µl PBS. The hemocyte suspension was transferred to a Neubauer-improved hemocytometer (Marienfeld) for counting under the microscope. Plasmatocytes and lamellocytes were classified based on their morphology. In total, more than 10 larvae were counted for each genotype. The Mann–Whitney test was used for the statistics.

### Quantitative PCR

Total RNA was prepared separately from third instar larval hemocytes, muscles, and the remaining corpse, by Aurum total RNA Mini Kit (Bio-Rad). Quantitative real-time PCR (qPCR) was performed using iScript One-Step RT–PCR Kit with SYBR Green (Bio-Rad), and the *RpL32* gene was used as a standard to normalize the RNA levels. Relative quantification of mRNA levels was calculated using Pfaffl's comparative cycle threshold ($C_t$) method that corrects for different reaction efficiencies [57]. The primers used and their reaction efficiencies are listed in Appendix Table S1. The normalized $C_t$ values do not distribute significantly different from the normal distribution (as tested by the D'Agostino-Pearson omnibus normality test, $P = 0.4946$, $n = 73$) and we could therefore use the *t*-test, with Welch's correction for unequal variances, for significance testing. The statistics was calculated on the $C_t$ values before transforming to the linear representation.

**Expanded View** for this article is available online:
http://embor.embopress.org

### Acknowledgements
We thank Bruno Lemaitre, Huaqi Jiang, Martin Zeidler, Nic Tapon, James Castelli-Gair Hombría, Ruth Palmer, Robert Schulz, and the Bloomington *Drosophila* Stock Center, for stocks. This research was supported by grants from the Swedish Research Council, the Swedish Cancer Society, the Academy of Finland, and the Sigrid Juselius Foundation.

### Author contributions
HY, JK, JE, and DH designed research; HY, GGK and JK performed research; HY, JK, JE, and DH analyzed data; and HY and DH wrote the paper.

### Conflict of interest
The authors declare that they have no conflict of interest.

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
