## [Review Process File · EMBO Reports]

Manuscript EMBOR-2015-40277

JAK/STAT signaling in *Drosophila* muscles controls the immune response against parasitoid infection

Hairu Yang, Jesper Kronhamn, Jens-Ola Ekström, Gül Gizem Korkut and Dan Hultmark

Corresponding author: Dan Hultmark, Umeå University

Review timeline:

Submission date:	24 February 2015
Editorial Decision:	25 March 2015
Revision received:	27 June 2015
Editorial Decision:	20 July 2015
Revision received:	19 August 2015
Editorial Decision:	20 August 2015
Revision received:	27 August 2015
Accepted:	28 August 2015

Transaction Report:

Editor: Nonia Pariente

1st Editorial Decision

25 March 2015

Thank you for your submission to EMBO reports. We have now received reports from the three referees that were asked to evaluate your study, which can be found at the end of this email. As you will see, all the referees find the topic of interest and in principle suitable for us, although referees 1 and 2 feel the study requires a number of important additional controls, experiments, statistical analyses and discussions to make the data conclusive. Both have indicated that significant revision would be required to strengthen the study sufficiently for publication here.

Given that all referees provide constructive suggestions on how to strengthen the work, I would like to give you the opportunity to revise your manuscript. If the referee concerns can be adequately addressed, we would be happy to accept your manuscript for publication.

All issues raised by referees 1 and 2 are reasonable and pertinent, and should be addressed. Although not essential for publication, the work would also benefit from taking into consideration the concern raised by referee 3 regarding negative data.

Please note that it is EMBO reports policy to undergo one round of revision only and, thus, acceptance of your study will depend on the outcome of the next, final round of peer-review.

Revised manuscripts must be submitted within three months of a request for revision unless previously discussed with the editor; they will otherwise be treated as new submissions. Please note that EMBO reports also publishes full-length articles now, and thus it would be possible to include more than five main figures, if necessary.

I look forward to seeing a revised form of your manuscript when it is ready. In the meantime, please contact me if I can be of any assistance.

REFEREE REPORTS

Referee #1:

In this article, the authors report that JAK-STAT signaling appears to be induced in muscles after infection by a parasitoid wasp. Two JAK-STAT ligands are induced in hemocytes, and their ectopic expression in hemocytes is sufficient to trigger JAK-STAT expression in muscles as well as to release lamellocytes in circulation. While the overall argument is rather convincing, some important controls are missing to validate this work.

The authors rely only on reporter transgenes with a low ratio of signal/noise (see more below) and do not analyze endogenous gene expression. Thus, the activation of the JAK-STAT pathway should be monitored by measuring SOCS36E expression in dissected larvae (carcass without the digestive tract, lymph glands, dorsal vessel and fat body, which are easy to dissect away in larvae).

The authors count hemocytes from bled larvae. The use of molecular markers such as antibodies would reinforce their argument by confirming their counts. Also, it would be necessary to check whether lamellocytes do not differentiate or do differentiate but adhere to tissues when JAK-STAT signaling is impaired in the muscles and thus are not released in circulation. This can be achieved by using antibodies. It would also be important to document the effect of blocking JAK-STAT signaling in the muscles on the lymph gland after infection.

There is a problem with statistical analysis for all experiments, including RT-qPCR analysis, except possibly the encapsulation data. This reviewer doubts that $p < 0.001$ in Fig. 1C. Indeed, the error bars (SD) overlap... More fundamentally, the authors use a parametric test, yet do not establish that the conditions required to use it are fulfilled, that is a normal distribution. To show this, current tests require at least 30 data points. Thus, the authors should use a nonparametric test and actually display each experimental value and not means or SD.

To activate the JAK-STAT pathway efficiently, it would be more efficient to target by RNAi a negative regulator of the JAK-STAT pathways. In any case, the ectopic activation of the JAK-STAT pathway should be documented using either the reporter transgene or RT-qPCR.

It is essential to show that Mef2 in larvae is expressed only in muscles (the expression of the driver has been analyzed in embryos). Given the recently described role of Mef2 in the systemic immune response, one would like to be sure that this gene is also not expressed in the larval fat body, as is the case in adults. Using a second independent driver expressed in the muscles is required to comfort the authors' conclusions that muscles represent the relevant tissue.

Minor points:

1. How often have the experiments been independently repeated? This is never stated.
2. What is the evidence that the p38b DN and bsk DN transgenes are working as expected in hemocytes?
3. A rescue experiment of either the Delta upd2 or Delta upd3 mutants by a transgene expressing one of the JAK/STAT ligands in hemocytes would strengthen the conclusions.
4. It is not entirely clear whether the 10x-STAT reporter transgene is expressed only in the muscles (Fig. 1). Crossing UAS-GFP with a somatic muscle driver would give a good reference point. Is the reporter induced in the dorsal vessel, in the lymph glands, other tissues: have the larvae been dissected?
5. Fig. 2E: a positive control, that is wild-type infected larvae, would provide the reader with an estimation of the strength of the transgenes effects.
6. Fig. 2J: a negative uninfected control would have been nice, although it is not essential here.
7. These remarks also apply to Figs 3J-N.
8. What is the efficiency of the primers couples? Do the authors fulfill the conditions to use the

comparative Ct method?

Referee #2:

This interesting paper documents a striking activation of STAT in *Drosophila* muscles following wasp infection, and shows that this is due to cytokine signalling from hemocytes. Stat activity in muscle also appears to be important for immunity against these infections, though why is not so clear. For the most part the experiments are elegant, the paper contains a number of striking results that are clearly meaningful, and the role of muscle in immunity is quite novel. Overall the report should be of interest to the insect immunity field and perhaps immunologists and physiologists in general. There are however a few important controls that need to be done, and a couple of obvious experiments are missing that might improve the paper if included. These are listed below, and the authors should respond with revisions. Most importantly, a specific function for muscle in mediating immunity is not proposed. The authors thoughts on this should be included in the abstract and discussion.

Specific comments:

1. The experiments shown in Figs 2 & 3 depend upon the specificity of the HML-Gal4 driver to hemocytes. If this driver is expressed in other cell types this would severely confound the authors' interpretation. Although this is a widely used hemocyte driver the authors should nevertheless document that it does not express in muscle.
2. Part of the authors' model is that the Upd2 & Upd3 cytokines are produced by hemocytes. However they test these cytokines using mutants that delete expression in all cells. Hemocyte-specific knock-downs (with RNAi) or ablation of hemocytes would help to validate their model. If possible, the authors should present such data, in addition to the mutant data, which is excellent.
3. On p 8 the authors overexpress WT STAT92E in muscles but find no effect. However this is not surprising as overexpression of STAT is not known to increase STAT activity. An activated JAK allele is available, however (hopTL) that should do the trick.
4. The discussion should include a proposal for what the specific role of the muscle might be in immunity against wasp infection. The paper contains no speculation about the meaning of this central finding.

Referee #3:

This is an interesting and novel manuscript. The authors use state of the art techniques to implicate body wall muscles as part of a signaling relay system in the *Drosophila* immune response to injection of a wasp egg. I don't have any issues that need to be addressed.

I find it difficult to conclude that a pathway isn't involved in a phenotype the basis of a negative result, say the failure of expression of a gene in a hemocyte to give a phenotype; for example, expression of a dominant negative domeless doesn't alter the response seen in these flies. There are many reasons this could happen. A trivial problem could be that the protein was never expressed. Probably that isn't happening because expression in another tissue had an effect, but how can we be sure. I think it is a fool's errand to keep chasing this down to ensure that the protein is expressed, which is to say I wouldn't hold a paper back for not testing this but it is a formal possibility. A more complicated issue could have to do with the dynamics of the response. Where is the response occurring on the domeless dose response curve? If the signaling response it is way out on the asymptote of maximum activation then you would require an enormous reduction in domeless to see an effect. I don't have some number in my head that says a gene has been knocked down enough to be sure this problem has been eradicated. This is a recurring problem in *Drosophila* immunity where we treat immune responses as if they were binary and not quantitative traits. Again, I don't expect the authors to chase this down and it wouldn't be fair because most fly immunity papers have this problem. My point is that it is hard to argue from a negative result to conclude that something doesn't happen.

I don't think there is any such thing as a standard *Drosophila* diet. Could the authors please provide the recipe?

Referee #1:

The authors rely only on reporter transgenes with a low ratio of signal/noise (see more below) and do not analyze endogenous gene expression. Thus, the activation of the JAK-STAT pathway should be monitored by measuring SOCS36E expression in dissected larvae (carcass without the digestive tract, lymph glands, dorsal vessel and fat body, which are easy to dissect away in larvae).

We have now also assayed JAK/STAT activity by measuring *Soc36E* expression in muscles. The results are shown in Fig. 1D, and they fully support the results obtained with the *10xSTAT-GFP* reporter.

The authors count hemocytes from bled larvae. The use of molecular markers such as antibodies would reinforce their argument by confirming their counts. Also, it would be necessary to check whether lamellocytes do not differentiate or do differentiate but adhere to tissues when JAK-STAT signaling is impaired in the muscles and thus are not released in circulation. This can be achieved by using antibodies.

We have now used a molecular marker, the MSN:Cherry lamellocyte reporter, to confirm that the number of lamellocytes is reduced when JAK/STAT signaling is suppressed (Appendix Figure S6). Furthermore, that experiment clearly demonstrates that the total number of lamellocytes in the animal is reduced, rather than redistributed between the sessile and circulating hemocyte populations.

It would also be important to document the effect of blocking JAK-STAT signaling in the muscles on the lymph gland after infection.

This is an interesting question, though somewhat beside the main scope of this manuscript. We have so far not obtained convincing results about such effects on the lymph gland. We have therefore left this part out.

There is a problem with statistical analysis for all experiments, including RT-qPCR analysis, except possibly the encapsulation data. This reviewer doubts that $p < 0.001$ in Fig. 1C. Indeed, the error bars (SD) overlap... More fundamentally, the authors use a parametric test, yet do not establish that the conditions required to use it are fulfilled, that is a normal distribution. To show this, current tests require at least 30 data points. Thus, the authors should use a nonparametric test and actually display each experimental value and not means or SD.

We agree with this criticism and we have now repeated the quantifications with larger numbers of animals. For the fluorescence experiments we have now also subtracted the background fluorescence of each microscopic field, thus increasing the signal to noise ratio. We have also tested whether the data fit a normal distribution. In cases when that criterion was not fulfilled we used a nonparametric test.

To activate the JAK-STAT pathway efficiently, it would be more efficient to target by RNAi a negative regulator of the JAK-STAT pathways. In any case, the ectopic activation of the JAK-STAT pathway should be documented using either the reporter transgene or RT-qPCR.

The efficient induction of JAK/STAT signaling in muscles by the *Mef2* driver is documented in Appendix Figure S5, using the *10xSTAT-GFP* reporter.

It is essential to show that Mef2 in larvae is expressed only in muscles (the expression of the driver has been analyzed in embryos). Given the recently described role of Mef2 in the systemic immune response, one would like to be sure that this gene is also not expressed in the larval fat body, as is the case in adults. Using a second independent driver expressed in the muscles is required to comfort the authors' conclusions that muscles represent the relevant tissue.

We have now confirmed our results with a second muscle-specific driver, *Twist-GAL4* (Fig. 5E). The *Mef2-GAL4* and *Twist-GAL4* drivers are well established in the literature. In addition we have also supplied our own data about their tissue specificity (Appendix Figure S4).

Minor points:

1. How often have the experiments been independently repeated? This is never stated.

Information about the number of animals assayed and the number of independent repeats are now indicated in the figure legends or in the figures.

2. What is the evidence that the p38b DN and bsk DN transgenes are working as expected in hemocytes?

Information about the efficiency of these constructs is given in the cited references and elsewhere. We have also specifically tested their specificity in hemocytes, but these data are included in a manuscript in preparation, with a different author constellation, and we prefer not to show them here.

3. A rescue experiment of either the Delta upd2 or Delta upd3 mutants by a transgene expressing one of the JAK/STAT ligands in hemocytes would strengthen the conclusions.

That is true but we have not been able to do that experiment in time for the revision. We hope that the weight of our other experiments is sufficient to convince the readers.

4. It is not entirely clear whether the 10x-STAT reporter transgene is expressed only in the muscles (Fig. 1). Crossing UAS-GFP with a somatic muscle driver would give a good reference point. Is the reporter induced in the dorsal vessel, in the lymph glands, other tissues: have the larvae been dissected?

We have dissected the animals and, as stated, the main part of the induction of the *10xSTAT-GFP* reporter is seen in muscles. Expression is seen in the lymph glands but we have not investigated this organ in detail. Background fluorescence is seen also in other tissues, but the only tissue where we can observe induced JAK/STAT signaling, besides the muscles, is the fat body where weak localized fluorescence is sometimes observed (Appendix Figure S1).

5. Fig. 2E: a positive control, that is wild-type infected larvae, would provide the reader with an estimation of the strength of the transgenes effects.
6. Fig. 2J: a negative uninfected control would have been nice, although it is not essential here.

These comments are true, but our other experiments provide the reader with sufficient information to estimate the effect. For instance, compare the relative induction in Fig 2E to the one shown in Fig 1 C. In both cases the experimental groups were normalized to the respective negative control group. So they are comparable.

7. These remarks also apply to Figs 3J-N.

In that case we do agree that a positive control for the infected animals is vital to show. Such positive controls were indeed part of the original experiments and we have included the data here (Fig. 3K and O). The p values in Fig. 3O are now, logically, related to that positive control.

8. What is the efficiency of the primers couples? Do the authors fulfill the conditions to use the comparative Ct method?

The PCR efficiencies for the different primer pairs are now indicated in Appendix Table 1. All quantitative PCR data are now recalculated and corrected for efficiency. No conclusions were affected by this additional data processing.

Referee #2:

1. The experiments shown in Figs 2 & 3 depend upon the specificity of the HML-Gal4 driver to hemocytes. If this driver is expressed in other cell types this would severely confound the authors' interpretation. Although this is a widely used hemocyte driver the authors should nevertheless document that it does not express in muscle.

The tissue specificity of the Hml-Gal4 driver is now illustrated in Appendix Figure 2E, where the bands of sessile hemocytes are clearly seen. No expression can be detected in the muscles.

2. Part of the authors' model is that the *Upd2* & *Upd3* cytokines are produced by hemocytes. However they test these cytokines using mutants the delete expression in all cells. Hemocyte-specific knockdowns (with RNAi) or ablation of hemocytes would help to validate there model. If possible, the authors should present such data, in addition to the mutant data, which is excellent.

We have now carried out that control experiment and the result supports our conclusions (Appendix Figure S3). The experiment shown in Appendix Figure 2, with the *upd1* and *upd3* reporters, did not suggest that any of these genes are induced in other tissues.

3. On p 8 the authors overexpress WT *STAT92E* in muscles but find no effect. However this is not surprising as overexpression of *STAT* is not known to increase *STAT* activity. An activated *JAK* allele is available, however (*hopTL*) that should do the trick.

Overexpression of wild-type *Stat92E* with the *Mef2* driver does in fact efficiently activate the $10\times$ *STAT-GFP* reporter in the muscles, as we now show in Appendix Figure S5. Overexpression of *hop*^{Turn} with this driver is unfortunately lethal.

4. The discussion should include a proposal for what the specific role of the muscle might be in immunity against wasp infection. The paper contains no speculation about the meaning of this central finding.

Exactly how the muscles exert their effect on the cellular immune defense, and why muscles of all tissues should play that role, remains a mystery. We simply don't know, although we are trying to find out. Reluctantly, we have added a few words of speculation on this point.

Referee #3:

*I find it difficult to conclude that a pathway isn't involved in a phenotype the basis of a negative result, say the failure of expression of a gene in a hemocyte to give a phenotype; for example, expression of a dominant negative domeless doesn't alter the response seen in these flies. There are many reasons this could happen. A trivial problem could be that the protein was never expressed. Probably that isn't happening because expression in another tissue had an effect, but how can we be sure. I think it is a fool's errand to keep chasing this down to ensure that the protein is expressed, which is to say I wouldn't hold a paper back for not testing this but it is a formal possibility. A more complicated issue could have to do with the dynamics of the response. Where is the response occurring on the domeless dose response curve? If the signaling response it is way out on the asymptote of maximum activation then you would require an enormous reduction in domeless to see an effect. I don't have some number in my head that says a gene has been knocked down enough to be sure this problem has been eradicated. This is a recurring problem in *Drosophila* immunity where we treat immune responses as if they were binary and not quantitative traits. Again, I don't expect the authors to chase this down and it wouldn't be fair because most fly immunity papers have this problem. My point is that it is hard to argue from a negative result to conclude that something doesn't happen.*

Yes, there can be many reasons for a negative result, although we and others have shown that the drivers and dominant negative constructs we tested are efficient in other combinations. The activated JAK/STAT signaling we observed in response to wasp infection could of course be important in ways that we don't detect with our encapsulation assay, but we refrain from speculations on this point.

*I don't think there is any such thing as a standard *Drosophila* diet. Could the authors please provide the recipe?*

Indeed, that is true. We now provide the recipe in the Appendix Supplementary Methods.

2nd Editorial Decision

20 July 2015

Many thanks for submitting the revised version of your manuscript to our editorial office. Your manuscript was sent back to two of the original referees and we have now received their feedback.

As you will see, while they in general appreciate the way in which you have addressed their initial

concerns, they also both still raise some issues that require additional clarifications, but also some experimental evidence.

Based on these reports, I would like to give you the exceptional opportunity to revise your study again along the lines indicated in their report.

Please do not hesitate to contact me if you have any further questions.

REFEREE REPORTS:

Referee #1:

The revised version addresses most of the comments this reviewer expressed. The article might have been somewhat stronger had the authors quantified endogenous JAK-STAT pathway activation in Fig. 2 and 3, in as much as the results shown in Fig. 1D are at the limit of the statistical significance convention.

Minor points:

- 1- Is there a statistically significant difference between STAT92E and dome DN transgenes in Fig. 5D? If it were the case, then the authors might want to explain why these constructs display the same encapsulation effect (Fig. 5A)
- 2- Discussion: when discussing JAK-STAT pathway induction in the visceral midgut muscles, the authors should cite not only the Jiang study but also a contemporary study from the Lemaitre laboratory.

Referee #2:

In this revision the authors have addressed my points #1, 3, & 4 well. They've also addressed most of comments from Reviewers 1 and 3 in reasonable ways. However my point #2 as not addressed in a satisfactory fashion. This is, specifically:

" Part of the authors' model is that the Upd2 & Upd3 cytokines are produced by hemocytes.

However

they test these cytokines using mutants the delete expression in all cells. Hemocyte-specific knockdowns

(with RNAi) or ablation of hemocytes would help to validate there model. If possible, the authors should present such data, in addition to the mutant data, which is excellent. "

The new data shown in Fig S2, regarding the expression of Upd2 and Upd3, are not clear or conclusive (upd1>GFP, Upd3>GFP). Better data should be offered to show that these cytokines are specifically expressed only in Hemocytes. In addition, the authors have not tried depleting Upd3 or Upd2 specifically in hemocytes with RNAi. This last experiment is essential, since it is the only test that could show specifically that hemocyte-derived Upd2 & Upd3 are important in triggering Stat activity in the muscle. The Upd2,3 deletion mutant data is suggestive but not conclusive.

2nd Revision - authors' response

19 August 2015

Referee #1:

The article might have been somewhat stronger had the authors quantified endogenous JAK-STAT pathway activation in Fig. 2 and 3, in as much as the results shown in Fig. 1D are at the limit of the statistical significance convention.

The control with an endogenous target gene for JAK/STAT signaling, *Socs36E*, was previously requested by this referee. We made that control, confirming our results with the *10x-Stat-GFP* reporter, but the referee is rightly pointing out that our control showed only borderline statistical significance. Indeed, we had only repeated the *Socs36E* assay in

five independent experiments and the magnitude of the induction was rather variable, explaining the poor statistical significance. We have now made five additional independent repeats of the same experiment with similar results, though this time with slightly lower variance. Our combined results are now included in the revised version of Fig. 1D. With a very high statistical significance ($p=0.00071$), they confirm that infection activates the JAK/STAT pathway, and that the magnitude of induction is similar (about twofold) when assayed with *10x-Stat-GFP* or *Socs36E*. Thus, we are confident that the *10x-Stat-GFP* reporter is a reliable indicator of JAK/STAT activity. It should therefore not be necessary to repeat also the experiments in Fig 2 and 3 by quantification of *Socs36E*. To do so within the given time frame would anyway be beyond our capacity.

Furthermore, we want to stress that the increased JAK/STAT activity in somatic muscles fits nicely with the increased expression in the infected animal of the cytokine genes *upd2* and *upd3*, and with our observation that suppression of the JAK/STAT pathway in muscle cells reduced the encapsulation of wasp eggs.

Minor points:

1- Is there a statistically significant difference between STAT92E and dome DN transgenes in Fig. 5D?

If it were the case, then the authors might want to explain why these constructs display the same encapsulation effect (Fig. 5A).

There is indeed a significant difference between the effects of the *Stat92EDN* and *domeDN* constructs on lamellocyte formation (Fig 5D), but not on encapsulation (Fig 5A). However, both lamellocyte formation and encapsulation are likely to be highly non-linear. Quantification of lamellocytes in circulation is particularly unreliable as a quantitative readout of the response, since activated lamellocytes tend to form nodules. Therefore, an initially strong lamellocyte response is sometimes followed by a drastic drop in the number of freely circulating lamellocytes. For these reasons we prefer to refrain from speculations on this point.

2- Discussion: when discussing JAK-STAT pathway induction in the visceral midgut muscles, the authors should cite not only the Jiang study but also a contemporary study from the Lemaitre laboratory.

The omission of that reference was unfortunate. It is now included in our revised manuscript.

Referee #2:

In this revision the authors have addressed my points #1, 3, & 4 well. They've also addressed most of comments from Reviewers 1 and 3 in reasonable ways. However my point #2 as not addressed in a satisfactory fashion. This is, specifically: "Part of the authors' model is that the Upd2 & Upd3 cytokines are produced by hemocytes. However they test these cytokines using mutants the delete expression in all cells. Hemocyte-specific knockdowns (with RNAi) or ablation of hemocytes would help to validate there model. If possible, the authors should present such data, in addition to the mutant data, which is excellent." The new data shown in Fig S2, regarding the expression of Upd2 and Upd3, are not clear or conclusive (upd1>GFP, Upd3>GFP). Better data should be offered to show that these cytokines are specifically expressed only in Hemocytes. In addition, the authors have not tried depleting Upd3 or Upd2 specifically in hemocytes with RNAi. This last experiment is essential, since it is the only test that could show specifically that hemocyte-derived Upd2 & Upd3 are important in triggering Stat activity in the muscle. The Upd2,3 deletion mutant data is suggestive but not conclusive.

We now see that our previous rebuttal letter was not entirely clear on this point. We do agree with this referee's point #2, and we had actually done the requested critical experiment. Using hemocyte-specific knockdowns (with RNAi), as suggested by the referee, we had shown that expression of the Upd2 & Upd3 cytokines in hemocytes is specifically required for the JAK/STAT response. The results are shown in Fig S3, not Fig S2.

We can't claim that these cytokines are expressed in hemocytes only, but our qPCR data (Fig 3E-F) as well as the GFP reporter data (Fig 3A-D) clearly show that only hemocytes show a strong induction of these cytokines in response to wasp infection. The data shown

in Fig S2 were not added in response to this referee's criticism. They were there already in the first version of our manuscript. The reporters we used in that experiment are admittedly rather weak, but the results do give further support to our claims.

3rd Editorial Decision

20 August 2015

Thank you for the submission of your revised manuscript to EMBO reports. Barbara Pauly contacted you with a decision on the previous version due to my absence from the office at that time. I have now assessed your answers to the previous referee reports and the modifications made to the study and I am happy to write with an 'accept in principle' decision, which means that we will accept your manuscript for publication once a minor editorial issue has been addressed, as follows:

- Your manuscript will be published as a full-length article, which means that additional figures can be accommodated, as necessary. I think it would be good to include at least SF3 into the main body of the manuscript.

After the issue has been attended to, you will receive an official decision letter from the journal accepting your manuscript for publication in the next available issue of EMBO reports. This letter will also include details of the further steps you need to take for the prompt inclusion of your manuscript in our next available issue.

Thank you for your contribution to EMBO reports. It has been a pleasure to handle your study and I look forward to seeing it published.

3rd Revision - authors' response

27 August 2015

We are very happy that you decided to accept "in principle" our manuscript *JAK/STAT Signaling in Drosophila Muscles Controls the Cellular Immune Response Against Parasitoid Infection*.

We have now followed Dr. Pariente's suggestion to add additional figure material from the Appendix to the main text. We have split Fig 3 in two, and included former Figure S3 as panel J in what is now Fig 4. We have also included former Figure S6 as panels F and G in what is now Fig 6. We have also revised figure references and captions in the main text. The affected text and figure files are now uploaded on your web site.

4th Editorial Decision

28 August 2015

I am very pleased to accept your manuscript for publication in the next available issue of EMBO reports.

Thank you again for your contribution to EMBO reports and congratulations on a successful publication. Please consider us again in the future for your most exciting work.